# Work-Related Injuries Reported to Workers Compensation Fund in Tanzania from 2016 to 2019

**DOI:** 10.3390/ijerph18179152

**Published:** 2021-08-30

**Authors:** Brenda S. Shewiyo, Hussein H. Mwanga, Ezra J. Mrema, Simon H. Mamuya

**Affiliations:** Department of Environmental and Occupational Health, School of Public Health and Social Sciences, Muhimbili University of Health and Allied Sciences, Dar es Salaam 11103, Tanzania; mwangahh@gmail.com (H.H.M.); ezrajm@yahoo.com (E.J.M.); mamuyasimon2@gmail.com (S.H.M.)

**Keywords:** work-related injuries, workers’ compensation, occupational health, workplace injuries in Tanzania

## Abstract

Quality data on the magnitude and determinants of work-related injuries is an important element in the development of appropriate preventative strategies. However, there have been many challenges in obtaining quality information on work-related injuries in the developing countries. This archival study based on the data from workers’ compensation registry from the year 2016 to 2019 aimed at determining trends and factors associated with work-related injuries (WRI). Data from 4578 WRI claims reported to Workers Compensation Fund (WCF) in Tanzania were analyzed. As expected, this new workers’ compensation system had increasing participation from inception in 2016 through 2019, resulting in increasing numbers of fatal and non-fatal work-related claims. Motor traffic accidents, machine faults and falls were the most reported causes of WRI. Males had more than 2-fold increased odds of sustaining fatal injuries compared to females. More than 6-fold increased odds of fatal injuries were observed for injuries occurring during conveyance. Explosions, motor traffic accidents, and falls were more likely to result into fatal injuries. Increased odds of fatal injuries were observed in workers from transportation and storage sector; information and technology; construction and building; and electricity, gas, and steam sectors, as well as among teachers, drivers, office workers, and security guards. The current study offers some insights regarding trends and associated factors that are vital in planning and implementation of appropriate preventative strategies for work-related injuries in Tanzania.

## 1. Introduction

Every day, thousands of workers die globally, and a hundred thousand suffer permanent disabling injuries, due to work-related accidents [1,2], with low income countries being the most affected [3]. In 2015, it was estimated that, in the Southern Africa region alone, 18,000 workers die from work-related accidents, more than 13 million are injured, and 67,000 contract occupational diseases [4]. In 2013, the National Audit Office of Tanzania (NAO) reported 121 fatal injuries from different sectors; construction and building was the leading industry, followed by transport and mining sectors [5]. Higher rates of occupational injuries were reported among Tanzanite small-scale miners in Tanzania [6].

Reducing the burden of work-related death is a major challenge since countries with the highest number of deaths are the ones with the poorest inspection and reporting systems [2]. While there is no comprehensive national surveillance system for occupational injuries in Tanzania, as in many other developing countries, important information can be obtained from the newly established Workers Compensation Fund (WCF). Workers compensation data contains valuable information commonly used in injury characterization [7] and have been recognized as a major source of workplace health and safety surveillance [8]. The information obtained from compensation data can provide a comprehensive understanding of injury patterns and further analyse factors associated with occupational fatalities [7]. However, compensation systems are burdened with a number of flaws, including under-reporting of claims, under-documentation of workers information, and poor record management [9].

The Tanzanian WCF was established in July 2015 but only started receiving workers compensation claims in July 2016. The scheme receives notifications from registered workers in both public and private sectors on the work-related injuries, diseases, and deaths for compensation purposes. The WCF data has limited information on both individual and work-related factors. Exposure information is mainly based on the job title and employment sector and lack important covariates, such as smoking, alcohol consumption, and level of education. Correctly documented compensation claims can be used as an important source of data to evaluate effects of occupational exposures [10,11].

Studies have shown that fatal consequences of injuries are strongly associated with both socio-demographic and work-related factors [12,13], so factors, such as age and sex, can influence the nature and extent of injuries since younger and male workers tend to be engaged in higher risks jobs [13,14,15]. In different occupations, teenagers and young adults have shown a higher risk of work-related injuries and fatalities [16], and men are three times more likely to experience severe injuries compared to women [17]. It has also been reported that an increase in education level alone cannot reduce accident rates in high risk workplaces [14,18]. Occupation (job type) was found to be strongly associated with severity of injuries and fatal outcomes [19].

In Tanzania, several workers are undoubtedly injured every day, but, due to the lack of centralized notification system, the majority of these injuries goes unnoticed. Little is known regarding the burden of work-related injuries in Tanzania. This pilot study aimed at identifying trends and factors associated with work-related injuries reported to WCF. Understanding trends and important predictors for work-related injuries (WRI) is important for policy and effective planning, with the overall aim of promoting and protecting workers’ health [20].

## 2. Materials and Methods

This was an archival study, analysing data from registries of the Tanzanian WCF from July 2016 to December 2019. Data collection and analysis started in January 2020. Incidents of work-related injuries, diseases, and fatalities are notified to WCF by workers for compensation purposes. A total of 4891 WRI were reported to WCF during this period. Among these, 313 cases were excluded due to lack of important information regarding the WRI (Figure 1). Among those excluded 4 cases were fatal reports from the year 2016 which had no information on the cause of death, while the remaining 309 were non-fatal injuries which had no information on the type, cause, and nature of injuries.

These data were collected by using a standard tool, adapted, and modified from the standard ILO occupational health surveillance data collection methods [1,21].

The independent variables of interest included socio-demographic factors (age, sex, and marital status), injury-related factors (location of injury, time of accident, cause of injury, and body part injured), and work-related factors (job title and employment sector). The main outcome variables included fatal and non-fatal injuries.

The data analysis was conducted by using STATA statistical software version 12 (StataCorp, College Station, TX, USA). Descriptive statistics were performed to obtain rates of occurrence of both fatal and non-fatal injuries in different groups. Chi-squared test was used to assess the correlation between socio-demographic variables and work-related injuries. Multivariate logistic regression models adjusted for age, gender, and marital status were used to assess the association between injury-related and work-related risk factors with the occurrence of work-related injuries. The dependent variable for multivariate logistic regression (work-related injuries) was a binary outcome comparing fatal versus non-fatal injuries. With the exception of age (continuous variable), all other variables were entered in the multivariate logistic regression models as categorical variables. A 95% confidence interval (CI) and a *p*-value of less than 0.05 was considered statistically significant for the association between work-related injuries and independent variables (such as cause of injury, time of incident, and body part injured). Potential confounders (age, sex, and marital status) were selected on an a priori basis, based on biological plausibility. Independent variables of interest (injury-related and work-related) were analyzed separately for each model, while adjusting for potential confounders. There was no power of estimation since this is a pilot study.

Ethical approval was obtained from the Senate Research and Publications Committee of Muhimbili University of Health and Allied Sciences (IRB#: MUHAS-REC-04-2020-255). Permission to use the WCF data was obtained from the Director General of the WCF. Confidentiality was ensured during the data abstraction and analysis. The names of the workers were removed from the database.

## 3. Results

A total of 4578 workers sustained work-related injuries that were reported to WCF from 2016 to 2019, of which 236 (5%) were fatal. The majority of workers who reported incidences of work-related injuries to WCF between 2016 to 2019 were men (83%), married (68%), and with a median age of 34 years (interquartile range (QR): 27–44 years) (Table 1). Further analysis showed that males had more than 2-fold increased odds of sustaining fatal injuries as compared to females (Unadjusted OR = 2.5; 95% CI: 1.6–4.1).

### 3.1. Trends in Work-Related Injuries Reported to WCF from 2016 to 2019

As expected, this new workers’ compensation system had increasing participation from inception in 2016 through 2019, resulting in increasing numbers of fatal and non-fatal work-related claims (Figure 2). The observed increasing trends in work-related injury claims were similar when comparing fatal against non-fatal injuries, as both had, on average, a 4-fold increase across the 4-year period.

### 3.2. Factors Associated with Work-Related Injuries Reported to WCF from 2016 to 2019

Tanzanian WCF compensates for injuries that have occurred in the workplace, as well as those occurring during conveyance of an employee to or from work. Although most injuries reported to WCF occurred in the workplace (66%), commuting injuries (34%) accounted for almost three-quarter (73%) of all fatal injuries (*p* < 0.001). Specifically, more than 6-fold increased odds of fatal injuries (adjusted odds ratio (AOR) = 6.1; 95% CI: 4.5–8.3) were observed for injuries occurring during conveyance (Table 2). The majority of the reported WRI (76%) occurred during day time.

Motor traffic accidents were the leading cause of both fatal (73%) and non-fatal WRI (31%). Explosions, motor traffic accidents, and falls were more likely to result into fatal injuries. Apart from motor traffic accidents, machine faults and falls were the most reported causes of WRI. Injuries involving multiple body parts and head trauma were shown to be more fatal than injuries involving other body parts (Table 2). When motor traffic accidents were excluded in the analysis (Table 3), no significant changes were observed with regard to the relationship between WRI and the predictor variables, such as time of incident, cause of the injury, and body part injured.

Compared to workers from the manufacturing sector, increased odds of fatalities were found among workers from the following sectors: transportation and storage; administrative and support; information and technology; accommodation, food, and beverage; professional, scientific, and technical; and construction and building (Table 4). When motor traffic accidents were excluded in the analysis, increased odds of fatal injuries were also observed in workers from electricity, gas, and steam (AOR = 5.0; 95% CI: 1.3–19) and wholesale and retail trade (AOR = 4.3; 95% CI: 1.1–17) as compared to workers from manufacturing sector (Table 5).

Teachers, drivers and office workers had increased odds of fatal injuries as compared to mechanics (Table 4). When motor traffic accidents were excluded in the analysis, increased odds of fatal injuries were also observed among security guards as compared to mechanics (Table 5).

## 4. Discussion

In this study, males had more than 2-fold increased odds of sustaining fatal injuries as compared to females. Furthermore, more than 6-fold increased odds of fatal injuries were observed for injuries occurring during conveyance. Motor traffic accidents, machine faults, and falls were the most reported causes of WRI. Explosions, motor traffic accidents, and falls were more likely to result into fatal injuries. Increased odds of fatal injuries were observed in workers from transportation and storage sector; administrative and support; information and technology; accommodation, food, and beverage; construction and building; electricity, gas, and steam; and wholesale and retail trade sectors, as well as among teachers, drivers, office workers, and security guards.

The increasing trends in work-related injuries observed in this study was most likely due to the increased registration of workplaces with WCF and increasing awareness among workers and employers regarding occupational health and workers’ compensation. Information obtained from WCF revealed that the number of workplaces registered with WCF increased exponentially during the period 2016 to 2019. There was no significant increase in the number of WRI in the observed years when data analysis was restricted to the workplaces that were registered in 2016.

The injuries reported had several different causes, among which explosions and motor traffic accidents were the leading causes of fatality. These findings were similar to the study done among Tanzanite mine workers in Tanzania, which also revealed that most frequent causes of fatality were falls and machine faults [6]. Studies conducted in Gambia, China, South Korea, and the United States also highlighted that falls from height were the leading cause of fatal injuries at work, followed by being struck by machine part or moving objects [22,23].

It is widely known that male workers are at higher risk of sustaining occupational injuries [24,25]. In the current study, male workers had more than 2-fold increased odds of sustaining fatal injuries as compared to female workers. These findings are similar to those reported in Australia, whereby men were found to have more physical injury claims than women, which were mostly attributed to occupational factors [26] since men tend to take higher risk jobs than female [6]. Other similar findings were reported by studies from Mexico and the UK, which reported higher risks of fatal injuries among males [13,27].

In the current study, higher odds of fatal injuries were observed for injuries occurring during commuting compared to injuries occurring at the workplaces. A study done in Malaysia had similar findings, which observed that, while industrial accidents were on the decline, commuting accidents had been rising to double their initial rates [28]. Other similar studies demonstrated that majority of fatal accidents at work were attributed to motor traffic accidents during commuting [20,29]. However a study done in Chile had contrary findings, whereby majority of fatal injuries occurred at workplaces and not during commuting [30]. The higher prevalence of injuries occurring during conveyance observed in the current study underscores the importance of improving road safety, along with other measures of safeguarding workers’ health and safety, in the overall efforts of protecting the working population.

Several studies have investigated the relationship between employment sectors and WRI and have identified employment sector as an important risk factor for the occurrence of fatal injuries in the workplaces [13,20,31,32]. In these studies, the agriculture, forestry, and fishing sector was found to be the leading industry in reporting fatal injuries among workers, followed by construction and mining sectors [23,25,32,33,34]. When conveyance injuries were excluded in the current study, workers from information and technology; electricity, gas, and steam; wholesale and retail trade; and construction and building sectors demonstrated higher rates of fatal injuries. The majority of fatal injuries were caused by electric shocks and explosions in electricity, gas, and steam; and information and technology sectors.

Job title/occupation is one of the important factors than the type of industry in explaining job risk pattern for injuries occurring in the workplaces [3,14,35]. In this current study, teachers, drivers, and office workers showed higher odds of fatal injuries when conveyance injuries were included. When conveyance injuries were excluded, higher odds of fatal injuries were observed among security guards and office workers (from agriculture and forestry sector, who died from animal attacks and violent human attacks). Higher odds of fatal injuries among drivers is not surprising since almost three-quarters of all fatal injuries reported were due to motor traffic accidents, but a high rate of motor traffic accidents among teachers and office workers is quite surprising. This can be partly explained by the fewer numbers of total registered workers in these job types. Similar results were reported in a French study that demonstrated that public workers had high road risk exposure close to that of manual workers [36]. Appropriate road safety measures need to be implemented to reduce the higher prevalence of fatal motor traffic injuries observed in the present study among teachers, office workers, and other public workers.

The findings in this study have provided an insight of the work-related injuries that typically occur among workers in the mainland Tanzania. Since higher rates of fatalities were observed for conveyance injuries, road safety measures need to be properly planned and implemented. For workplace injuries, regular workplace hazards identification and health risk assessment should be conducted, followed by the implementation of appropriate preventive and control measures to ensure protection and maintenance of workers’ health and safety. Furthermore, due to the relatively high number of fatalities caused by electric shocks and explosion, regular preventive maintenance of the electrical systems is imperative in the workplaces.

This is the first study to have conducted an analysis of the WCF data in Tanzania. However, there are some limitations in our study that need to be considered. The current study’s findings are relevant to only formal workplaces in the mainland Tanzania that were registered with WCF during this period (2016–2019). Moreover, it is most probable that not all WRI that occurred during this period were reported to WCF. This could have resulted in selection bias if reported WRI were significantly different from those that were not reported. Furthermore, there were under documentation/missing information of the reported cases for example only 40% of the reported injuries had information about location of the injury.

## 5. Conclusions

This study demonstrated increasing trends in both fatal and non-fatal WRI reported to WCF from the year 2016 to 2019. Increased odds of sustaining fatal injuries were observed among male workers as compared to female workers. Furthermore, injuries occurring during conveyance were associated with more than 6-fold increased odds of fatal injuries compared to fatal injuries that occurred in the workplace. Motor traffic accidents, machine faults, and falls were the most reported causes of WRI. Injuries resulting from explosions, motor traffic accidents, and falls were more likely to be fatal. Increased odds of fatal injuries were observed in workers from the transportation and storage sector; administrative and support; information and technology; accommodation, food, and beverage; professional, scientific, and technical; construction and building; electricity, gas, and steam; and wholesale and retail trade sectors, as well as among teachers, drivers, office workers, and security guards. More studies need to be conducted to further explore the important risk factors for WRI in the African context so as to enable the development of the most appropriate preventative strategies for this important public health problem.

## Figures and Tables

**Figure 1 ijerph-18-09152-f001:**
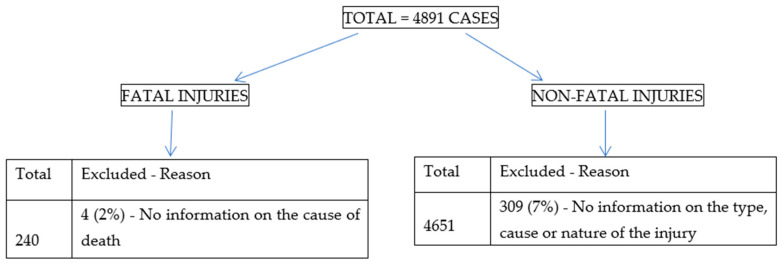
Distribution of work-related injuries reported to WCF during the study period.

**Figure 2 ijerph-18-09152-f002:**
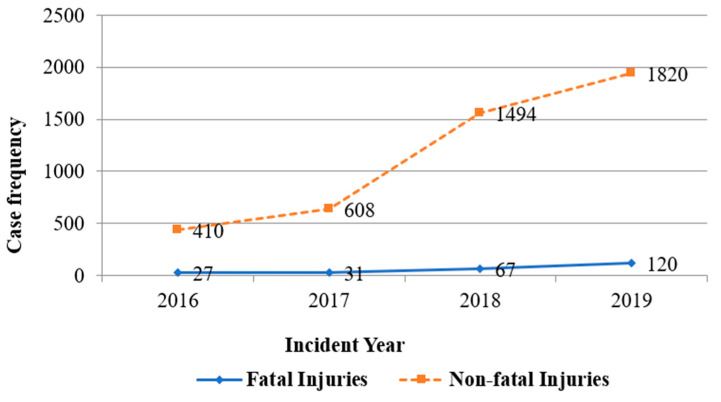
Trends in work-related injuries reported to WCF from 2016 to 2019.

**Table 1 ijerph-18-09152-t001:** Socio-demographic characteristics of workers who sustained work-related injuries.

Characteristics	Overall *n* (%)	Non-Fatal Injuries *n* (%)	Fatal Injuries *n* (%)	*p*-Value (Chi-Squared Test)
Age (years)	*n* = 4393	*n* = 4157	*n* = 236	
Median (Interquartile Range)	34 (27–44)	35 (29–42)	34 (27–44)	
<31	1862 (42)	1778 (42)	84 (36)	0.025
31–49	1880 (43)	1759 (42)	121 (51)	
>50	651 (15)	620 (14)	31 (13)	
Sex	*n* = 4452	*n* = 4217	*n* = 234	<0.001
Females	749 (17)	731 (17)	18 (8)	
Males	3703 (83)	3486(83)	217 (92)	
Marital status	*n* = 3427	*n* = 3234	*n* = 193	0.126
Married	2314 (68)	3234 (67)	140 (73)	
Not married	1113 (32)	1060 (32)	53 (27)	

**Table 2 ijerph-18-09152-t002:** Factors associated with work-related injuries reported to WCF from 2016 to 2019.

Variable	Overall *n* (%)	Fatal Injuries *n* (%)	Non-Fatal Injuries *n* (%)	AOR (95%CI)
Location of injury	*n* = 2856	*n* = 231	*n* = 2625	
Conveyance	979 (34)	169 (73)	810 (31)	6.1 (4.5–8.3) *
Work place	1877 (66)	62 (27)	1815 (69)	ref
Time of accident	*n* = 4392	*n* = 96	*n* = 4295	
Day time	3359 (76)	76 (79)	3282 (76)	1.2 (0.7–1.9)
Night time	1033 (24)	20 (21)	1013 (24)	ref
Cause of Injury	*n* = 2848	*n* = 231	*n* = 2617	
Animal/human attack	131 (5)	10 (4)	121 (5)	4.3 (0.9–20.0)
Burn	106 (4)	2 (1)	104 (4)	ref
Explosions	80 (3)	15 (7)	65 (2)	12.0 (2.7–54.0) *
Fall	587 (21)	14 (6)	573 (22)	1.3 (2.7–5.7) *
Hand tools faults	185 (6)	0 (0)	185 (7)	1 (0.2–8.4)
Machine faults	785 (27)	21 (9)	764 (29)	1.4 (0.3–6.2)
Motor traffic accidents during conveyance	974 (34)	169 (73)	805 (31)	10.9 (2.7–44.7) *
Body part injured	*n* = 3848	*n* = 164	*n* = 3684	
Head	423 (11)	33 (20)	389 (11)	0.7 (0.4–1.5)
Upper extremities	1551 (40)	3 (2)	1543 (42)	0.02 (0.0–0.01) *
Lower extremities	867 (23)	1 (1)	865 (23)	0.01 (0.0–0.1) *
Trunk	127 (3)	12 (7)	103 (3)	ref
Multiple injuries	899 (23)	115 (70)	784 (21)	1.3 (0.7–2.4)

AOR = adjusted odds ratio (adjusted for age, gender and marital status); ref = reference group; Day time: 06:00–18:59; Night time: 19:00–05:59; * *p*-value < 0.05.

**Table 3 ijerph-18-09152-t003:** Factors associated with work-related injuries that occurred exclusively in the workplace (i.e., excluding motor traffic accidents).

Variable	Overall *n* (%)	Fatal Injuries *n* (%)	Non-Fatal Injuries *n* (%)	AOR (95%CI)
Time of the incident	*n* = 3517	*n* = 27	*n* = 3490	
Day time	2696 (77)	21 (78)	2675 (77)	1.1(0.4–2.7)
Night time	821 (23)	6 (22)	815 (23)	ref
Cause of Injury	*n* = 1874	*n* = 62	*n* = 1812	
Animal / human attack	131 (7)	10 (16)	121 (7)	4.3 (0.9–20.0)
Burn	106 (6)	2 (3)	104 (6)	ref
Explosions	80 (4)	15 (24)	65 (4)	12.0 (2.6–54.0) *
Fall	587 (31)	14 (23)	573 (31)	1.3 (0.3–5.7)
Hand tools faults	185 (10)	0 (0)	185 (10)	1.0 (0.2–8.4)
Machine faults	785 (42)	21 (34)	764 (42)	1.4 (0.3–6.2)
Body part injured	*n* = 3086	*n* = 50	*n* = 3036	
Head	341 (11)	12 (24)	329 (10)	0.3 (0.1–0.6) *
Upper extremities	1439 (47)	2 (4)	1437 (47)	0.01 (0.0–0.4) *
Lower extremities	694 (22)	0 (0)	694 (23)	1.0 (0.02–3.9)
Trunk	72 (2)	9 (8)	63 (2)	ref
Multiple body part injuries	540 (18)	27 (54)	513 (17)	0.4 (0.2–0.8) *

AOR = adjusted odds ratio (adjusted for age, gender and marital status); ref = reference group; Day time: 06:00–18:59; Night time: 19:00–05:59; * *p*-value < 0.05.

**Table 4 ijerph-18-09152-t004:** Work-related injuries reported to WCF classified by employment sector and job title.

Variable	Overall *n* (%)	Fatal Injuries *n* (%)	Non-Fatal Injuries *n* (%)	AOR (95%CI)
Employment Sector	*n* = 4554	*n* = 243	*n* = 4311	
Accommodation, Food and beverage	131 (3)	8 (3)	123 (3)	5.0 (1.7–19) *
Administrative and Support	502 (11)	51 (21)	451 (10)	5.5 (3.0–13.8) *
Agriculture and Forestry	905 (20)	28 (12)	877 (20)	1.5 (0.6–4.0)
Construction and building	390 (9)	27 (11)	363 (8)	4.1 (1.6–10.5) *
Electricity, Gas, and steam	185 (4)	12 (5)	173 (4)	2.8 (0.1–8.4)
Financial and Insurance services	72 (2)	4 (2)	68 (2)	2.0 (0.4–8.7)
Human Health and Social	198 (4)	4 (2)	194 (5)	1.0 (0.2–4.2)
Information and Technology	84 (2)	11 (5)	73 (2)	5.1 (1.5–16.6) *
Manufacturing	1077 (24)	16 (7)	1061 (25)	ref
Mining and Quarrying	57 (1)	4 (1)	53 (1)	1.2 (0.1–11)
Professional, Scientific and technical	234 (5)	13 (5)	221 (5)	4.2 (1.4–12) *
Security Groups	200 (4)	8 (3)	192 (4)	1.2 (0.3–4.3)
Transportation and Storage	314 (7)	37 (15)	277 (6)	6.4 (2.5–17) *
Wholesale and Retail trade	205 (5)	20 (8)	185 (4)	3.7 (1.3–11)
Job title	*n* = 3520	*n* = 207	*n* = 3313	
Administrator	166 (5)	15 (7)	151 (5)	1 (0.3–3.1)
Driver	341 (10)	58 (28)	283 (9)	3.4 (1.5–8.0) *
Engineer	86(2)	9(4)	77(2)	1.1 (0.2–5.6)
Security Guard	279 (8)	18(9)	261 (8)	1.2 (0.5–3.1)
Health worker	142 (4)	6 (3)	136 (4)	1.2 (0.3–4.3)
Labourer	564 (16)	26 (13)	538 (16)	1.3 (0.6–3.1)
Mechanic	343 (10)	8 (4)	335 (10)	ref
Office worker	232 (7)	22 (11)	210 (6)	3.4 (1.3–8.9) *
Operator	877 (25)	17 (8)	860 (26)	0.4 (0.2–1.3)
Teacher	86 (2)	10 (5)	76 (2)	5.1 (1.7–15) *
Technician	376 (11)	18 (9)	358 (11)	1.4 (0.5–3.5)

AOR = adjusted odds ratio (adjusted for age, gender and marital status); ref = reference group; * *p*-value < 0.05.

**Table 5 ijerph-18-09152-t005:** Work-related injuries (excluding motor traffic accidents) reported to WCF classified by employment sector and job title.

Variable	Overall *n* (%)	Fatal Injuries *n* (%)	Non-Fatal Injuries *n* (%)	AOR (95%CI)
Employment Sector	*n* = 3581	*n* = 72	*n* = 3509	
Accommodation, Food and beverage	110 (3)	0 (0)	110 (3)	1.0 (0.4–22)
Administrative and Support	304 (9)	5 (9)	299 (7)	2.7 (0.7–11)
Agriculture and Forestry	743 (21)	10 (21)	733 (14)	1.5 (0.5–4.7)
Construction and building	319 (9)	14 (9)	305 (19)	3.8 (1.3–11) *
Electricity, Gas, and steam	133 (4)	7 (4)	126 (10)	5.0 (1.3–19) *
Financial and Insurance services	41 (1)	1 (1)	40 (1)	1.1 (0.2–17)
Human Health and Social	147 (4)	1(4)	146 (1)	1.0 (0.3–9.2)
Information and Technology	59 (1)	4 (2)	55 (6)	6.0 (1.3–27.5) *
Manufacturing	1012 (29)	10 (28)	1002 (14)	ref
Mining and Quarrying	46 (1)	3 (4)	43 (1)	1.2 (0.1–14)
Professional, Scientific and technical	191 (5)	4 (5)	187 (6)	3.5 (0.9–13.5)
Security Groups	118 (3)	2 (3)	116 (3)	1.8 (0.6–12)
Transportation and Storage	209 (4)	4 (6)	205 (6)	1.6 (0.3–8.4)
Wholesale and Retail trade	149 (4)	7 (4)	142 (10)	4.3 (1.1–17) *
Job title	*n* = 2613	*n* = 58	*n* = 2555	
Administrator	98 (4)	4 (7)	94 (4)	2.3 (0.2–26)
Driver	175 (7)	7 (12)	168 (6)	8.6 (1.5–51) *
Engineer	65 (2)	3 (5)	62 (2)	2.9 (0.2–33)
Security Guard	192 (8)	7 (12)	185 (7)	5.6 (1.1–32) *
Health worker	99 (5)	1 (2)	98 (4)	6.0 (0.5–77)
Labourer	448 (16)	12 (21)	436 (16)	4.3 (0.9–20)
Mechanic	315 (13)	2 (3)	313 (12)	ref
Office worker	150 (4)	6 (10)	144 (5)	24.7 (3.7–164) *
Operator	716 (28)	6 (10)	710 (29)	1.0 (0.2–9.8)
Teacher	42 (2)	0 (0)	42 (2)	0.01 (0.0–15)
Technician	313 (11)	10 (17)	303 (11)	4.1 (0.8–21)

AOR = adjusted odds ratio (adjusted for age, gender and marital status); ref = reference group; * *p*-value < 0.05.

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
