# Peer review of "Work-Related Injuries Reported toWorkers Compensation Fund in Tanzania from 2016 to 2019"

_ijerph, 2021, doi:10.3390/ijerph18179152_

Round 1

Reviewer 1 Report

I would like to thank the authors for revising the manuscript and responding to the comments raised in the previous review cycle. While the quality and organization of the paper improved, additional work is needed to ensure the value of the article is clearly communicated to future readers.

Abstract –

  • Add a sentence at the end of the abstract to highlight the value of the insight generated (how it contributes to public health and safety).

Introduction -

  • Consider moving paragraph in Line 53 – 59 before Line 47.
  • This section is still scanty and should be strengthened significantly. I commented on the paucity of information in the previous round, and I will be unable to recommend an “Accept” until the authors provide a clear and convincing argument for this study. For instance, the authors should provide a summary of studies that utilized similar archival methods to extract trends from compensation data. The authors should provide an additional paragraph that summarizes the problem/gap and need for this study. This information is not clear in the present structure. This should come before Line 59 - “The aim of this study…”
  • Socio-demographic, injury-related, and job-related factors were assessed as independent variables. Provide a paragraph explaining/summarizing previous studies that utilized these variables. The current version simply glances over this important aspect.
  • Ensure proper and effective transitions between paragraphs in this section.

Methods -

The authors indicate that there was no power estimation because the study was a pilot study. There was no reference to the study being a pilot study upstream. The authors should include this information in the abstract and introduction section.

Results - 

Figure 1 and Line 117-119 indicate an increasing trend. However, the authors state that there was no increase after controlling for the increased WCF registration (Line 184 – 188). Line 190 also states that there was an increase in injuries/fatalities as well. Kindly reconcile these conflicts.

Discussion - 

While the authors improved the content in the discussion section, there remain opportunities for improvement. The authors should provide interpretations (the relevance of the result and how the information could be used in Tanzania or other developing countries). For instance, results from the study indicate that teachers, drivers, office workers, and security guards have higher rates of reporting fatal injuries. This result should inform an action. Are there studies that have provided interventions for improving the safety of these workers? The researchers should provide possible actions that the government or organizations could implement to improve the safety of workers. Each paragraph in the discussion section should provide relevant take way (potential solutions and actions) from the analysis

Overall, the quality of the paper improved significantly. However, the paper will benefit from another round of revision to improve the quality of the content and. Also, some I become additional editing to improve flow and eliminate grammatical mistakes.  

Reviewer 2 Report

This is a very important and well-written manuscript.  The following suggestions have been made to address minor points that may improve clarity.

1.  For the sentence on Lines 32 - 34 that begins with "It is estimated that in Southern Africa. . . ", it would be helpful to indicate the time period that the estimates cover.

2.  In the Introduction, it would be informative to discuss the reason that marital status, mentioned on Line 79, is included as an independent variable, and why it is thought to be a confounder, as stated in Line 93.   

3.  Can the authors discuss potential reasons that increased odds of fatal injuries were found in seemingly low-risk sectors including administrative and support, information technology, and professional, scientific, and technical (Lines 177 - 179).  

4.  Can the authors discuss potential reasons that teachers and office workers have higher rates of reporting fatal injuries (Lines 218 and 219)?

5.  It would be helpful to point out additional data fields that would be useful for WCF to collect to aid in identifying work-related risk factors.  For example, having the number of full-time workers per company per year, or the total number of hours employees in a company worked per year, would be immensely helpful for calculating injury rates and enabling a comparison of injury risk across sectors.

Reviewer 3 Report

  1. In line-70-77, the flow chart lacks a title. Also, the flow charts content "Exlcuded and Reason" should be put in the same column.
  2. In line 85, the word “association” could be revised to “correlation.”
  3. In Table 1, the analysis is only for chi-square and descriptive method. However, in line 107-107, the result of Table 1 shows that males had more than 2-fold increased odds of sustaining fatal injuries as compared to females. This paper should clearly explain the results of Table 1 and give the results of odds ratio in Table 1.
  4. In Table 1, the numbers (4393, 4157, 236) should put in the row of "Age".
  5. In line 102, the total of 4578 is correction or not because the calculation result is 4568. Calculation: 4881-4-309=4568.
  6. In line 127, how to get the value (p<0.001) is. This value is for "odds ratio" or from other statistic method.
  7. In the discussion section, figure 1 must be discussed because it shows the important result. The lines for fatal injuries and non-fatal injuries from 2016 to 2019 have the big change. What is the reason for the big change?
  8. In Table 1, most fatal injuries are from motor traffic. Hence, this paper should explore deeply the reasons.
  9. The preventive strategies for fatal injuries and non-fatal injuries for Tanzania should be explored in the analysis results.

Reviewer 4 Report

This paper describes a retrospective examination of workers’ compensation (WC) records obtained from the Tanzanian WC Fund; a national data collection program that became effective in July 2016. Records analyzed included claims from the beginning through 2019. Analyses included reports of work fatalities and work injuries. The authors compared three categories of factors: socio-demographic factors, personal factors, and job factors. I do not consider the paper in its present form suitable for IJERPH. I offer several suggestions the authors may want to consider for improving the paper and submitting to a different journal or perhaps to a scientific conference that publishes a proceeding.

Abstract

Lines 20-21, the sentence giving specific reference to software belongs in the body of the paper, not in the Abstract.

Lines 21-22. I don’t think it is correct to say “The study demonstrated increasing trends in work-related injuries from 2016 to 2019.” This could mislead readers into believing there is a trend over time for work injuries and fatalities in Tanzania to be climbing. In reality, the number of reports being collected by the record system has been growing. Lines 23 to 28 were presented in a kind of zig-zag order, starting with fatalities, (sex and conveyance), then work-related injuries (motor traffic, machine faults, and falls), then back to fatalities (explosions, motor traffic accidents, and falls).  

Key Words

The first three key words are good but he fourth “workers’ health” is not actually a topic in the paper.

Introduction

Lines 33: Suggest checking the statement about “higher rates” because I noticed the title of the paper referenced [2] is prevalence not rates.

Paragraph 1 starts on the topic of global injuries and fatalities. The second half of paragraph abruptly changes to the topic of experience in Tanzania. Then paragraph 2 goes back to global matters. The Introduction would flow better if you make all of paragraph 1 be about global matters, and bring Tanzania matters into paragraph 2.

Line 53. Is WCF part of the official name of the organization? Or is the organization name Tanzania Workers’ Compensation Fund?

Materials and Methods

I recommend being more explicit on the method used to get adjusted odds ratio. This measure is your number 1 measure in Tables 2, 3, and 4. Why not show equation used or explicitly describe the computational procedure ussed. A related suggestion is to put and example in a footnote to each of these tables showing how the AOR was computed for one of the factors in the table. Somewhere in the paper, you should state the reason for selecting particular reference groups for these AORs.

Results

There are multiple references to “increasing trends’ in this section. Authors might want to consider starting the section with something like “As expected, this new record system had increasing participation from inception in 2016 through 2019., resulting in increasing numbers of fatal and non-fatal work-related claims (Figure 1)”

Lines 137. Suggest revising paragraph 1 to something like the following.

Compared to workers from the manufacturing sector, increased odds off fatalities were found among workers from the following sectors: transportation and storage; administrative and support; information and technology; accommodation; food and beverage; scientific and technical; and construction and buildings  

Discussion

Lines 231-234: What is the point of the last sentence in the Discussion? Workers’ compensation records are only numerator data for risk. Without exposure information (denominator data), one cannot compute risk and cannot study personal risk factors like BMI. The really major thing you should mention is the  limitation of having only numerator data.

References

Numerous cited articles/reports do not include the publisher, or website, or Doi.

Round 2

Reviewer 3 Report

For the reviewer's previous point 9, the authors should add a section to discuss the preventive strategies in order to reduce work-related injuries according the analysis results. However, the authors still cannot supply the preventive strategies in here. Hence, this paper's contribution is only to analysis the data set. It should give a contribution, supplying some strategies, to reduce the injuries.

Author Response

For the reviewer's previous point 9, the authors should add a section to discuss the preventive strategies in order to reduce work-related injuries according the analysis results. However, the authors still cannot supply the preventive strategies in here. Hence, this paper's contribution is only to analysis the data set. It should give a contribution, supplying some strategies, to reduce the injuries.

Response: Thank you for the input, a paragraph has been added in the discussion to describe strategies to reduce the occurrence of work-related injuries.

Reviewer 4 Report

My review of the second submission of this paper is more favorable than the initial submission. I have some specific points for the authors, but each is easily fixable.

Line 45-47. The second part of the sentence is poorly worded. On line 47 is should read factors associated with occupational fatalities.

Line 56. WC data cannot be used to evaluate worker’s health. Recommend changing to evaluate effects of occupational exposures.

Sentence starting on line 128. I suggest: The observed increasing trends in work-related injury claims were similar when comparing fatal against non-fatal injuries….

Lines 227-230. The authors mention two sources of fatalities (electric shock and explosions). The next sentence contains recommendations of one of the two sources. Either delete the recommendation about electric shock, or add recommendations about explosions. Personally, I don’t see how recommendations for reducing risk fits into the aims and scope of this paper.

Lines 253-254. Consider extending the sentence: Furthermore, injuries occurring during conveying were associated with more than 6-fold increased odds of fatal injuries compared to fatal injuries that occurred in the workplace.

Author Response

This manuscript is a resubmission of an earlier submission. The following is a list of the peer review reports and author responses from that submission.

Round 1

Reviewer 1 Report

Dear authors,

thank you for the research. The study investigated work-related injuries in Tanzania. Using the Tanzanian Workers' Compensation Fund, the authors conducted association analyses between exposures (based on job titles and employment sector) and work-related injuries and calculated odds ratios to show the association. The English language of the article is ok but could be improved and described a bit more scientifically. The authors did not perform excessive self-citation.

Unfortunately, however, I found numerous inconsistencies in content and methodological deficiencies. These are, for example, inconsistent presentation of numbers in tables, missing information on author affiliation, insufficient author information, incorrect verbatim quotations, missing number of the ethics vote, different numbers in text and figures (although both describe the same facts), duplication of sentences in terms of content. In addition, I am critical of the model for association analyses because important and relevant confounders were not considered in this study.

In my opinion, important information about the data set, information for a possible replication of the study, and some literature references also seem to be missing or could be incomplete. The estimation of exposure based on job titles and employment sectors probably contributes strongly to a reality bias, which reduces the quality of the study. In addition, the graph is not properly labeled, and the introduction seems incomplete to me. I also think the introduction distorts the current study situation. Overall, I see an urgent need for revision, especially in the introduction, methodology, and results, which will inevitably lead to a change in the discussion. I wish the authors much success.

Kind regards

General comments:

Please check the sentence structures in the sentences with a reference statement throughout the entire manuscript. The reference statement should be written before the punctuation mark and with a space before the reference statement. Otherwise, it is grammatically incorrect. See, e.g., line 27: Suggested change: However, only a few studies have investigated this problem [1].

Tables 2, 3, 4, 5: Please define "ref" in the legend. Please ensure consistent spelling of AOR information in the tables, this includes spaces before brackets, specifying a decimal digit for all values, and describing in the legend that "bold" means significant.

Specific comments:

#1. Line 6 - 7: Institutions/companies/affiliations 1 to 4 are missing, please add according to the style of the journal IJERPH (at least institution, post code, city, country). Please use professional email addresses when possible. It is not possible to tell who the corresponding author is from the correspondence email address. Please choose a unique email address. It is best to add the name abbreviations after the e-mail addresses (e.g., mwangahh@gmail.com (H. H. M.)). Please check previously published articles in the journal IJERPH if you are unsure of the author information. I believe a telephone number should be included only for the corresponding author.

#2. Line 9: A colon missing after ABSTRACT.

#3. Line 10. Since you give 2.78 million as a concrete number, you are unfortunately forced to provide a literature reference in the abstract, since this number will not be referenced later in the manuscript. If this is not possible, please put your statement into relative perspective without hard facts, e.g., "Worldwide, many people still lose their lives during performing their daily work.“

# 4. Line 23 -24: Please remove the dot after the "keywords".

#5. Line 25: Please use a space between "1." and "INTRODUCTION".

#6. Line 27: "However, only a few studies..." here you give only one reference but report few studies. Please review this sentence and either write about only the one study or provide more reference information. Also, I believe many more studies are looking at this issue as it is a common issue - even in other countries. Please review the studies again and adjust this aspect. If you are referring to only a specific country or city with your statement, please adjust the sentence and add the restrictions.

#7. Line 28. There is a missing transition from a global view to a specific view in Africa. Please adjust the text.

#8. line 29: Please add a space after "13" and before "millions".

#9. Lines 28 - 34: Your reference [4] is from 1991. In my opinion, this is not a current source to describe a current topic. Your reference [2] from 2013 also seems relatively "old" to me. I believe there are more current numbers you should be referring to. Please use more current references to address the issue and justify why research in this area is required. To do this, please review the current studies in this field of research and adjust your introduction.

#10. Lines 33 - 34: Please cite the acquisitions correctly word for word. Please follow the style of the journal IJERPH. In addition to citing the source, please include at least the page number.

#11. Line 39 - 40: If you report "worldwide", I will ask you to provide more reference information. Possibly references from different countries, nations, or continents will be fine. One study on Korea is not enough to make a worldwide statement. Also, you have not yet addressed the issue of occupational cancer. Either stick with your examples (traffic accidents, machine errors and falls, ...) or adjust the introduction and possibly add cancer. The way the introduction is written now, the text passages do not fit together directly.

#12. Lines 44 - 45: The goal of your study is not entirely clear from the introduction. I would ask you to explain in the introduction why it is important for exactly Tanzania to conduct this research. It is also unclear to the reader why you chose this period (2016-2019) since it is currently 2021 and the WCF existed since 2015. In my opinion, a review of the dataset from 2015 (date close to the start of the WCF) to at least 01 July 2020 would be more appropriate. Otherwise, I do not see your research as current, which makes it difficult to justify this as current publication.

#13. Line 47: Reference to # 12: Please indicate the exact date from when to when you studied the dataset. Information on days and months is important for possible replication of your study. It seems to me that the analysis of the dataset is not up to date and incomplete. Logical would be to study WCF from 2015 (date close to operation of WCF) to at least 01 July 2020.

#14. lines 47 - 53: It would be more descriptive to show the exclusion of cases in a flow chart, including the exact definition of the exclusion criteria. It would also help the reader if you describe the WCF in more detail. What type of data are included? What type of diseases and complaints are included in the data? How are the exposures categorized? Are the exposures entered by title and Job Exposure Matrix, are they self-reported exposures, or were the exposures measured by sensors, cameras, or ….? This information is missing at this point. Where is this WCF stored? Is it publicly available? Does it contain sensitive data? Or similar...

#15. Line 54: Please insert the distributing company and country in STATA.

#16. Line 55: Please define exactly the variables, especially the dependent variables for your logistic regressions.

#17. Line 62 - 63: Please specify the number of your ethical vote.

#18. Line 47 - 64: Please provide information on the following: What statistical parameters do you want to calculate? What model did you use (including reference)? If you developed your own model, please provide a detailed description of the model for your regressions. Please indicate in the methodology that the model provides odds ratios. Please indicate how you subdivide the exposure for the analyses (continuous data, categorical/ordinal scaled data, or normal scaled)? Please provide information on a power estimation. Or alternatively, indicate that you are conducting a pilot study for which you do not necessarily need a power estimation. Please revise your methodology, especially the adjustment of the model (see comment #19).

#19. Lines 67 - 73: Please write before the "5% were fatal" the number of fatal injuries out of 229. Please provide additional information about your sample. What was the BMI of the workers? What was the level of physical/sporting activity? What was the percentage of smokers? Were these individuals more likely to drink alcohol or use comparable substances regularly? What was the level of job satisfaction of the workers? What is the degree of handedness in the sample (right-handed, left-handed, ambidextrous)? How many years did the subjects perform their job? What is the level of comorbidity? If there are other confounders, please think about to include them as well. Confounders such as smoking, sport, comorbidity, BMI can accelerate or increase the development of disease, discomfort, or fatal injury. In the discussion, you describe with reference [14] that alcohol abuse leads to a large proportion of fatal injuries. Why did you not adjust the model for this confounder if you already cite this study? Without adjusting for these factors, you can only provide limited conclusions with your correlation analyses.

I would ask you to adjust your model, include more factors in the adjustment, and recalculate your analyses. Adjusting for age, gender, and marital status is not sufficient, in my opinion, to make qualitative and reliable statements about the sample.

#20. Line 70: There is a missing space between "Table" and "1". Your model is adjusted for sex, age, and marital status. I think this model with adjustment is inappropriate for calculating the OR for fatal injuries. At least, the reader might think so. Please indicate here that the analysis is an unadjusted result. Because I do not think a reader will conclude this from the context.

#21. Line 71: Please provide the abbreviation for confidence interval already in line 59.

#Line 75: Your description in the last 4 years does not match the time of submission of your article, please adjust the text.

#23. line 78: In Figure 1 you indicate 2015 as year, but this is not included in the analyses. Please adjust the x-axis. Your Figure 1 draws attention to the non-fatal injuries because of the red color. The information that is important is lost. Please use dashed (-  -  -  -  - ) and colored lines at the same time, because the graph will probably be shown in black and white in a later possibly hardcover print. Otherwise, it will not be possible to match the legend. My recommendation is the following: Use a medium gray, dashed for the non-fatal injuries and a black line for the fatal injuries. Additionally, move the numbers slightly away from the line, otherwise they will be harder to read. Additionally, I ask for a heading/label for the x-axis and a heading/label for the y-axis.

Your numbers in Figure 1 do not match the numbers in the text. The summed fatal injuries give n = 245. In the text you give only 5%, which corresponds to n = 229. How can you explain this difference? The nonfatal injuries and the fatal injuries from Figure 1 add up to n = 4635. In the text, you describe a total of 4578. How do you explain the difference?

#24. Line 84: Please state the abbreviation adjusted odds ratio (AOR) at least once in full in the text.

#25. Line 100: Please swap the order of the text passages in the results. In the text, you described Table 4 first, then Table 3 which might be incorrect. Please use the same chronological order and labeling in the text and tables.

#26. Lines 96 - 107: In the text, you repeat the contents of the tables, especially the odds ratios. Please describe the results in a more interesting way without duplicating the tables in the results. I suggest you look at the style of results descriptions in other published articles.

#27. Line 140 - 141: The sentence is duplicate in content to the sentence in line 125 - 126. Please remove the duplicate sentence. In general, this information is very common in the article. Please check the entire article for redundancies.

#28. Line 142: Here you give a study from Tanzania with the reference [10]. But it was not given in the introduction. It therefore reinforces my suspicion that the introduction should be updated.

#29. Lines 202 - 249: Please check the references: [3], [4], [5], [8], [9], [16] for completeness, as e.g., the publisher, place, page number or similar are missing.

Reviewer 2 Report

I would like to thank the editor for this opportunity to review this manuscript that introduces an analysis on WRIs based on a Tanzanian accident database. I have written below my major concerns concerning this manuscript.

Introduction: the authors state as their first sentence that only a few studies have investigated trends and predictors for WRIs and base their claim on reference 1. I dare to question this. Is it really so in a global context? I really don’t think so. As the reference 1 seems to have studied factories in Addis Ababa, I really ask if they have concluded so. The rest of the Introduction sector seems ok. However, given the existing literature available in this topic in general, I consider that the authors should have made a proper literature review supplementing the Introduction. As such the Intro part with references is far too straight forward and limited. I mean that accident analyses on large statistics have been performed by various authors and this article would benefit getting familiar with those.

Materials and methods: The statistical part seems ok, however I miss reasoning for the indicators chosen. Are they selected on practical reasons, i.e. these are the ones the database includes, or is there some other reasoning behind?

Results: The authors first report findings related to men-dominance in the injuries studied. The problem in this is that they do not report how is this gender balance in general concerning the workforce in Tanzania and what does this mean from that perspective? Concerning figure 1, I’m wondering what does it tell us? Yes, the trend seems to be increasing and rather significantly, I must say, but why? Is it due to the maturity of the system, i.e. WRIs are reported better during the last years for some reason? So, to my understanding these should be referred to some other source, like conducted working hours in Tanzania (has it remained the same in 2016-2019) or something to be able to claim this increase. The rest of the results section seem easier to understand as in these the accidents are considered as a whole, not compared between years. However, when the authors state that increases on WRIs can be detected from various sectors is it really so or a result of the reporting system being more mature.

Discussion: Please see my comments concerning the results. They include aspects that should be considered in the Discussion sector as well. Other thing I’m missing is that this section is inadequate for a scientific article discussion. The authors should elaborate more in-depth how their results add scientific knowledge on this topic that they state in the beginning of this article to be studied only in a few studies. What new does this bring to discussion on WRI analysis from the methodological point of view, or does this study add something from the OSH science perspective. As now, it seems more a report paper with very little content from the science perspective. Concerning commuting there are various studies that have focused in this topic, please see for example a recent review: https://onlinelibrary.wiley.com/doi/full/10.1002/job.2462. Concerning the structure of the discussion, I find it unmatured. The authors would benefit structuring it in subchapters where scientific implications, managerial/practical issues, future studies and limitations are more in-depth presented.    

As a whole I see that the authors report some significant findings related to WRIs in Tanzania. However, the authors have not managed to build a scientific frame around this based on existing literature and the discussion around their findings is very unmatured. Therefore, I must regret and say that in its’ current form this manuscript is not acceptable to be published as a journal article in Int. J. Env. Res. & Public Health.

Reviewer 3 Report

The present study aims to determine trends and factors associated with work-related injuries (WRI) reported to WCF in Tanzania between 2016 and 2019. Literature on worker injury trends in Africa is scarce, so this study can contribute to occupational safety and health knowledge and practice in developing countries, especially those in Sub-Saharan Africa. However, the present study requires significant revision to provide the most value to future readers.

First, the Introduction section was shallow and lack a statement that properly delineates the need for this study. Moreover, there is no Background/Literature Review section. This section could summarize previous studies that discussed occupational injuries and fatalities trends. This would help show the gap in literature in Sub-Saharan Africa. Also, this section could provide a summary of studies that utilized similar methods research on trends (WCF). Finally, this section could describe WCF in Tanzania in more detail. The data reporting and collection process should be described here. Some limitations could be highlighted as well.  

Define WRI before using the acronym in the main body, not just in the abstract

The authors inaccurately labeled this study as a cross-sectional study. This study relied on archived data, and better fits the characteristics of an Archival Study.  http://pressbooks-dev.oer.hawaii.edu/psychology/chapter/approaches-to-research/

The references provided in Line 53 (8,9) to support the data collection tool used in WFC were not academic articles and lacked useful information to properly assess the data collection tool. I recommend that the authors provide an example of the tool used to collect Workers’ comp data.   

The sample size for certain characteristics is significantly less than the project sample size (e.g., Marital status is ~1,000 less than the project sample size).  How does this impact the analysis performed?

Section 3.1 has only one sentence. More information should be extracted from the trend. Was there a specific reason for the space in Page 4? Kindly delete the space if there is no reason.

The statement in line 83-85 could be confusing. Add 34% after the commuting injuries.

For a better appreciation of the result, it is important for readers to understand what the different accident types and characteristics entail. Is motor accident = conveyance or does motor accident include accidents that happen in a work-place involving a vehicle? For instance, construction is known to have a high number of vehicular injuries and fatalities due to worker – vehicular/equipment interaction while at work. Additional insight is needed.

It was strange seeing “information and technology”, “accommodation, food and beverages,” and “professional scientific and technical” sectors ranked higher than sectors such as construction, which are known to have very high worker injury and fatality statistics. The authors should provide a more detailed comparison of t this result, and other results, with existing literature. This could lead to interesting contributions to knowledge.

In Table 2, the sample size of “Location of injury” was almost 40% less than the study sample. This should lead to a biased conclusion. Address this in the revised document. Also, it was difficult to verify if the motor traffic accidents listed in this table were strictly conveyance or not.

According to Table 3, Administrative and Support sector contributed 21% of fatality injuries, which I find interesting. I believe this warrants a detailed discussion. How many companies in the WFC database were in this sector?  

In Table 4, does Multiple Injuries and Multiple body part mean the same thing? The article talks about injuries in multiple body parts, not multiple injuries. These are not synonymous since you could have multiple injuries on one body part.

Multiple numbers are incorrect in Table 5. Kindly review and fix.

The discussion is extremely shallow. There’s little contribution to knowledge or practice. I believe the authors can significantly improve this section.   

In Line 137 – 139, the authors mention that there was a significant increase in WRI when controlled for the rapid increase in registration. Does that mean that all references in the manuscript to increasing injury and fatality trends are in-accurate and should be deleted?

The reference used in Line 149 is for a different type of accident, and not a good citation for the sentence. Kindly revise.

The manuscript should be reviewed one more time for grammar. 

Overall, while I believe the study has some potential, the authors need to provide a deeper reflection in the discussion section.